# Substance use disorder among adolescents before and during the COVID-19 pandemic in Uganda: Retrospective findings from a psychiatric ward registry

Mark Mohan Kaggwa[1,2]*, Joan Abaatyo[1]*, Emmanuel Alol[1], Moses Muwanguzi[3], Sarah Maria Najjuka[4], Alain Favina[1], Godfrey Zari Rukundo[1,2], Scholastic Ashaba[1], Mohammed A. Mamun[5,6,7,8]

1 Department of Psychiatry, Faculty of Medicine, Mbarara University of Science and Technology, Mbarara, Uganda, 2 African Centre for Suicide Prevention and Research, Mbarara, Uganda, 3 Faculty of Medicine, Mbarara University of Science and Technology, Mbarara, Uganda, 4 College of Health Sciences, Makerere University, Kampala, Uganda, 5 CHINTA Research Bangladesh, Savar, Dhaka, Bangladesh, 6 Department of Public Health, University of South Asia, Dhaka, Bangladesh, 7 Department of Public Health, Daffodil International University, Dhaka, Bangladesh, 8 Department of Public Health and Informatics, Jahangirnagar University, Savar, Dhaka, Bangladesh

* Kmarkmohan@gmail.com (MMK); joandux@gmail.com (JA)

**Data Availability Statement:** All the data used is available from: https://doi.org/10.6084/m9.figshare.19411322.

## Abstract

It has been reported that the COVID-19 pandemic has predisposed adolescents to risky behaviors such as substance use and subsequent substance use disorder (SUD). However, it is unknown how the pandemic has changed the prevalence of SUD among adolescents in Uganda. We aimed to determine the prevalence of SUD and associated factors among adolescents in southwestern Uganda. Retrospectively, psychiatry ward records from November 2018 to July 2021 were collected from the largest tertiary hospital in southwestern Uganda. A total of 441 adolescent records were included in the analysis, with a mean age was 17 ±1.88 years, and the majority were males (50.34%). The overall prevalence of SUD was 7.26% (5.90% and 9.80% *before* and *during* the pandemic). Despite a little rise in SUD (3.9% increment) *during* the COVID-19 pandemic, there was no statistical difference compared to *before* the pandemic. The likelihood of being diagnosed with SUD was more among older adolescents at any period. In addition, having a diagnosis of bipolar mood disorder reduced the likelihood of SUD *during* the pandemic. This study indicated no statistical change in the diagnosis of SUD among adolescents before and *during* the COVID-19 pandemic. As older-male adolescents (17 to 19 years) were at higher risk of SUD, there is a need for early intervention for this group.

## Introduction

Adolescence (10–19 years) is characterized by a series of developmental changes, which are highly impacted by social, cultural, and nutritional influences [1]. A vast array of neurodevelopmental changes occur during this time, including cortical thinning, gray matter volume

**Funding:** The author(s) received no specific funding for this work.

**Competing interests:** The authors have declared that no competing interests exist.

reductions, increases in white matter volume, synaptic pruning, and reorganization within cortical and limbic regions [2–5]. These neurodevelopmental changes give rise to characteristic behaviors during adolescence, such as improvements in cognition and executive functions, increases in reward sensitivity, novelty-seeking, risk-taking behavior, and a tendency to spend more time with peers [6–8]. Some of these behavioral characteristics, in turn, contribute to a greater likelihood of initiating substance use [9, 10].

Substance use by adolescents remains a significant public health concern [1]. More than 50% of substance use initiation cases occur during adolescence [11, 12]. The common substance used among adolescents is alcohol, followed by marijuana and cigarette smoking [13, 14]. In the USA, cigarette smoking trends have been increasing due to the introduction of e-cigarettes, although cannabis is the most common substance used by adolescents [14]. In sub-Sahara Africa–a region with the highest population of adolescents, the overall prevalence of adolescent substance use was estimated at 41.6% [15]. A study in Kenya among pregnant adolescents (aged 14–18 years) found alcohol use disorder at 13.2% [16]. In Uganda, 60–71% of school-going children (12 to 24 years) use addictive substances, especially alcohol (19.3%) and Kuber (smokeless tobacco, used sublingually) at 4.4% [13]. In a study among adolescents attending the Makerere/Mulago Columbia Adolescent Health Clinic in Mulago, 15.6% used at least an addictive substance, with alcohol being dominant at 15.2% of the total population [17]. Moreover, an earlier age of onset of substance use is significantly associated with the risk of developing a substance use disorder (SUD) later in life [1, 18, 19]. Despite studies showing a prevalence of substance use among adolescents in Uganda, no particular study has been carried out to investigate the prevalence of SUD in the country.

Prior Ugandan studies reported that factors associated with substance use (as opposed to substance use disorder) among adolescents include the death of a mother, suffering from chronic illness, depression, and having friends or family members with substance use problems [16, 17]. However, following the COVID-19 pandemic, substance use and substance use disorders have been reported to have increased along with changes in associated factors [20, 21]. The factors leading to the aforementioned increase in substance use are due to the drastic change in the lifestyle of most individuals due to lockdown, social isolation, restriction of movement, and economic shutdown [20, 21]. The COVID-19 associated social and economic restrictions led to decreased opportunities for regulation of stress, anxiety, existing mental health conditions, domestic violence, child maltreatment, and traumatic experiences, hence paving the way for substance use among adolescent victims as a mode of coping [20–23]. Eventually, suicide occurrences were caused by the inaccessibility of alcohol due to lockdown, and alcohol withdrawal symptoms were reported during the pandemic [24, 25].

In Uganda, there is limited evidence on the impact of the COVID-19 pandemic on substance use disorder among the vast young population. Yet, the country had its schools closed for almost two years with no online studies or other means [26], thus removing this known protective factor (staying in school and school connectedness) against substance use among adolescents [27]. Furthermore, previous literature reports that adolescents out of school are at a higher risk of using addictive substances [28], thus putting them at risk of being diagnosed with SUD. Therefore, in this study, a retrospective record review was performed to compare the prevalence and associated factors of SUD based on two-time points, that is, *before* and *during* the COVID-19 pandemic among adolescents with mental health problems attending a psychiatric ward in Uganda.

## Methods

### Study area and design

This study was based on a retrospective review of the outpatient registers (Health Management Information Systems [HMIS] form 031) of the Mbarara Regional Referral Hospital (MRRH) psychiatry ward from November 2018 to July 2021. The psychiatry unit manages patients with mental illnesses confirmed by mental health professionals only and those with medical conditions that have mental illness presentations such as HIV. MRRH psychiatry ward is the largest psychiatric facility in southwestern Uganda, with a stationed child and adolescent psychiatrist and a functional specialist addiction clinic. Therefore, the facility handles and manages many adolescents especially referrals from all over southwestern Uganda, with mental illnesses, including SUD. *HMIS form 031* captures the following information; patient's number, name, address, age, anthropometric measurements, gender, next of kin, substances of addiction used, investigations, physical symptoms, diagnosis, and treatment given. At the beginning of the COVID-19 pandemic in Uganda (March 2020), 16 months data of *before* the COVID-19 pandemic and 17 months *during* the pandemic, were collected for this study.

### Inclusion and exclusion criteria

Using the *HMIS form 031* register, information of all adolescents aged 10 to 19 years [29] was extracted. After retrieving all the eligible registers from the psychiatry department records office, a total of 720 records were retrieved. Due to reattendances to the clinic, records with the same patient number, age, gender, patients' names initials, year of attendance, and diagnosis were considered duplicates, and only one record was included in this study. A total of 279 records were reattendances in the same calendar year for the same diagnosis, and so were excluded from the final analysis as duplicates.

### Data collection, management and quality control

The following information was extracted from the *HMIS form 031*: (i) patient's number, (ii) patient name initials, (iii) age, (iv) gender (male, female), (v) substances of addiction used (alcohol; cannabis; cigarettes (smoked tobacco), Kuber (smokeless tobacco, commonly used sublingually) [30]; miraa/khat [*Catha Edulis Forsk*]–plant leaves that contain stimulant similar to Amphetamine [31]; others such as Akandi [32], and glue), and (vi) diagnosis. Age was categorized into three groups based on the stage of adolescence [33]; early adolescents (10–14 years), middle adolescence (15–17 years), and late adolescence (18 to 19 years). Data were entered parallel by two pairs of individuals (JA and EA) and (SMN and CK), and in case of any discrepancies, MMK resolved them. Data were cleaned in Microsoft Excel.

### Substance use disorder and mental health diagnosis

The mental health diagnoses at the MRRH psychiatry ward are based on the ICD-11. Patients with physical diagnoses are referred to the department for management of their psychiatric manifestations or comorbidities. For this study, a diagnosis of SUD was considered for an individual who had been diagnosed with any type of SUD, such as alcohol use disorder, cannabis use disorder, among others [34]. An individual can be diagnosed with more than one type of substance use disorder. In addition, individuals with SUD can have both mental comorbidities such as depression, anxiety, or physical comorbidities such as HIV.

## Ethics

The present study was conducted in accordance with the Declaration of Helsinki 2013 [35] and was approved by the Mbarara University of Science and Technology research ethics committee (MUST-2021-229). However, the formal consent was waivered since the data was retrospective, and it was not possible to track the involved participants. All of the information is anonymously presented in this study and there are no ethical concerns.

## Data analysis

Data were analyzed using STATA version 12.0. Categorical variables were presented with frequencies and percentages. Age was presented in terms of mean and standard deviation. The Gaussian assumption was used to assess for normality based on the Shapiro-Wilks test and histograms. Chi-square tests for categorical variables or student *t*-tests for continuous variables were performed to determine significant differences between individuals with a diagnosis of SUD and those without. Inferential analysis of two groups, *before* and *during* the pandemic, was performed with the included study variables. Binary logistic analysis for factors associated with SUD was performed. Factors significant at bivariate logistic analysis were tested for collinearity using variance inflation factors (VIF), and those with a VIF below three were included in the final model at multiple logistic regression. The significant level was at less than 5% for a 95% confidence interval.

## Results

A total of 441 records were finally included in the present analysis, with 65.31% (n = 288) being recorded *before* the COVID-19 pandemic. The average age of the participants included was 17±1.88 years, with almost half (48.75%) being in their late adolescence (18–19 years). About half were males (50.34%), and the commonest psychiatric diagnosis made was bipolar disorder (40.59%), followed by schizophrenia and other primary psychosis (27.66%). Detailed information on adolescent data use in this study can be found in **Table 1**.

### Prevalence of substance abuse disorder and substance use types

The overall prevalence of SUD was 7.26% (n = 32), where 5.90% (n = 17) adolescents had SUD *before* the pandemic and 9.80% (n = 15) *during* the pandemic. Although a 3.9% increment of SUD *during* the COVID-19 pandemic was observed, the difference was not statistically significant ($\chi^2$ = 2.26, *p* = 0.133). The most used substance was cannabis (5.21%, n = 23/441), followed by alcohol, 4.76%, (n = 21/441), and the least used was Kuber 0.23% (n = 1/441). There was a statistically significant difference between cigarette use and the pandemic period (p = 0.023), although other substances were not significantly associated (**Fig 1**).

### Distribution of study variables with substance use disorder

The mean age of adolescents with SUD was higher compared to those without SUD (18.16 vs. 16.91 years; t = -3.66, *p*<0.001), that is, 12.56% of late adolescents had been diagnosed with SUD, whereas it was 2.73% for middle-aged adolescents and there was no SUD case in early adolescents, and this association of age group with SUD was statistically significant $\chi^2$ = 17.91, *p*<0.001). Furthermore, all adolescents with SUD were males (14.55% vs. 0%), and the association between gender and SUD was statistically significant ($\chi^2$ = 34.06, *p*<0.001). In addition, adolescents with schizophrenia and bipolar mood disorders were less likely to be diagnosed with SUD. For details about the relationship between the study variables and SUD, please refer to **Table 1**.

**Table 1. Distribution of the socio-demographics and comorbid mental and physical illnesses with substance abuse disorder.**

| Variable | Total; n (%) | Substance use disorder | | X² (p-value) |
|---|---|---|---|---|
| | | No; 409 (92.74%) | Yes; 32 (7.26%) | |
| **Socio-demographic information** | | | | |
| **Admission time** | | | | |
| Before pandemic | 288 (65.31) | 271 (94.10) | 17 (5.90) | 2.26 (0.133) |
| During pandemic | 153 (34.69) | 138 (90.20) | 15 (9.80) | |
| **Age (mean, *SD*)** | 17.00,1.88 | 16.91,1.90 | 18.16,1.02 | **-3.66 (<0.001)** |
| **Stage of adolescence** | | | | |
| Early | 43 (9.75) | 43 (100) | 0 | **17.91 (<0.001)** |
| Middle | 183 (41.50) | 178 (97.27) | 5 (2.73) | |
| Late | 215 (48.75) | 188 (87.44) | 27 (12.56) | |
| **Gender** | | | | |
| Female | 217 (49.66) | 217 (100) | 0 | **34.06 (<0.001)** |
| Male | 220 (50.34) | 188 (85.45) | 32 (14.55) | |
| **Comorbid mental illnesses and physical illnesses** | | | | |
| **1. Schizophrenia and other primary psychosis (6A2)** | | | | |
| No | 319 (72.34) | 288 (90.28) | 31 (9.72) | **10.38 (<0.001)** |
| Yes | 122 (27.66) | 121 (99.18) | 1 (0.82) | |
| **2. Bipolar mood disorders (6A8)** | | | | |
| No | 262 (59.41) | 232 (88.55) | 30 (11.45) | **16.87 (<0.001)** |
| Yes | 179 (40.59) | 177 (98.88) | 2 (1.12) | |
| **3. Depression** | | | | |
| No | 394 (89.34) | 363 (92.13) | 31 (7.87) | 0.36 (0.548) |
| Yes | 47 (10.66) | 46 (97.87) | 1 (2.13) | |
| **4. Anxiety and fear-related disorders (6B0)** | | | | |
| No | 426 (96.60) | 394 (92.49) | 32 (7.51) | 1.21 (0.270) |
| Yes | 15 (3.40) | (100) | 0 | |
| **5. Neurodevelopmental disorders (6A0)** | | | | |
| No | 431 (97.73) | 399 (92.58) | 32 (7.42) | 0.80 (0.371) |
| Yes | 10 (2.27) | 10 (100) | 0 | |
| **6. Factitious disorders (6D5)** | | | | |
| No | 435 (98.64) | 403 (92.64) | 32 (7.36) | 0.48 (0.490) |
| Yes | 6 (1.36) | 6 (100) | 0 | |
| **7. Elimination disorders (6C0)** | | | | |
| No | 440 | 408 (92.73) | 32 (7.27) | 0.08 (0.779) |
| Yes | 1 | 1 (100) | 0 | |
| **8. Personality disorder (8D1)** | | | | |
| No | 439 (99.55) | 407 (92.71) | 32 (7.29) | 0.16 (0.692) |
| Yes | 2 (0.45) | 2 (100) | 0 | |
| **9. Dissociative disorders (6B6)** | | | | |
| No | 440 (99.77) | 408 (92.73) | 32 (7.27) | 0.08 (0.779) |
| Yes | 1 (0.23) | 1 (100) | 0 | |
| **10. Disruptive behavior and dissocial disorders (6C9)** | | | | |
| No | 439 (99.55) | 407 (92.71) | 32 (7.29) | 0.16 (0.692) |
| Yes | 2 (0.45) | 2 (100) | 0 | |
| **11. Catatonia** | | | | |
| No | 440 (99.77) | 408 (92.73) | 32 (7.27) | 0.08 (0.779) |
| Yes | 1 (0.23) | 1 (100) | 0 | |

(*Continued*)

**Table 1.** (Continued)

| Variable | Total; n (%) | Substance use disorder | | X² (p-value) |
|---|---|---|---|---|
| | | No; 409 (92.74%) | Yes; 32 (7.26%) | |
| **12. Disorders specifically associated with stress disorders (6B4)** | | | | |
| No | 431 (97.73) | 400 (92.81) | 31 (7.19) | 0.11 (0.735) |
| Yes | 10 (2.27) | 9 (90.00) | 1 (10.00) | |
| **13. Mental and behavioral disorders associated with pregnancy, childbirth and the puerperium, (6E2)** | | | | |
| No | 440 (99.77) | 408 (92.73) | 32 (7.27) | 0.08 (0.779) |
| Yes | 1 (0.23) | 1 (100) | 0 | |
| **14. Disorders of bodily distress and bodily experience (6C2)** | | | | |
| No | 432 (97.96) | 400 (92.59) | 32 (7.41) | 0.72 (0.397) |
| Yes | 9 (2.04) | 9 (100) | 0 | |
| **15. Impulsive control disorders (6C7)** | | | | |
| No | 438 (99.32) | 406 (92.69) | 32 (7.31) | 0.24 (0.627) |
| Yes | 3 (0.68) | 3 (100) | 0 | |
| **16. HIV** | | | | |
| No | 428 (97.05) | 396 (92.52) | 32 (7.48) | 1.05 (0.306) |
| Yes | 13 (2.95) | 13 (100) | 0 | |

There was a statistical difference between the stage of adolescents and SUD *before* the COVID-19 pandemic ($\chi^2$ = 14.77, $p$ = 0.001), whereas no difference was observed *during* the pandemic. More males were statistically more diagnosed with SUD during the two periods ($p$<0.001); however, they were more prevalent *during* the pandemic than *before* the pandemic (18.29% vs. 12.32%). *Before* the pandemic, adolescents without comorbidity of schizophrenia and other primary psychosis (6A2) were less prone to SUD (8.13% vs. 0%, $\chi^2$ = 6.83, $p$ = 0.009); but such association was not statistically significant *during* the pandemic. Similarly, adolescents without comorbid bipolar disorder were more likely to be diagnosed with SUD in both periods (**Table 2**).

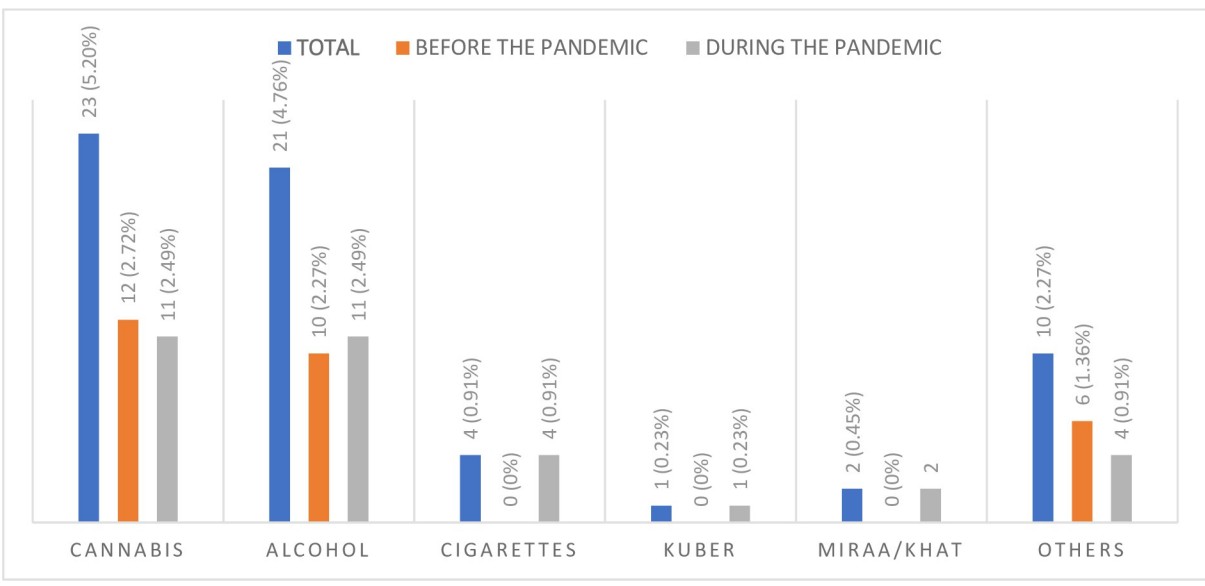

**Fig 1. Substances used *before* and *during* the COVID-19 pandemic.**

**Table 2. Distribution of the socio-demographics and comorbid mental and physical illnesses with substance abuse disorder based on *before* and *during* the pandemic.**

| Variables | Before COVID-19 pandemic | | | During the COVID-19 pandemic | | |
|---|---|---|---|---|---|---|
| | No | Yes | χ² (*p*-value) | No | Yes | χ² (*p*-value) |
| | (271, 94.10%) | (17, 5.90%) | | (138, 90.20%) | (15, 9.80%) | |
| **Stage of adolescence** | | | | | | |
| Early | 29 (100) | 0 | **14.77 (0.001)** | 14 (100) | 0 | 4.64 (0.098) |
| Middle | 117 (99.15) | 1 (0.85) | | 61 (93.85) | 4 (6.15) | |
| Late | 125 (88.65) | 16 (11.35) | | 63 (85.14) | 11 (14.86) | |
| **Gender** | | | | | | |
| Female | 148 (100) | 0 | **19.38 (<0.001)** | 69 (100) | 0 | **14.01 (<0.001)** |
| Male | 121 (87.68) | 17 (12.32) | | 67 (81.71) | 15 (18.29) | |
| **Comorbid mental illnesses and physical illnesses** | | | | | | |
| **1. Schizophrenia and other primary psychosis (6A2)** | | | | | | |
| No | 192 (91.87) | 17 (8.13) | **6.83 (0.009)** | 96 (87.27) | 14 (12.73) | 3.78 (0.052) |
| Yes | 79 (100) | 0 | | 42 (97.67) | 1 (2.33) | |
| **2. Bipolar mood disorders (6A8)** | | | | | | |
| No | 156 (90.17) | 17 (9.83) | **12.01 (0.001)** | 76 (85.39) | 13 (14.61) | **5.55 (0.018)** |
| Yes | 115 (100) | 0 | | 62 (96.88) | 2 (3.13) | |
| **3. Depression** | | | | | | |
| No | 239 (93.36) | 17 (6.64) | 2.26 (0.133) | 124 (89.86) | 14 (10.14) | |
| Yes | 32 (100) | 0 | | 14 (93.33) | 1 (6.67) | 0.18 (0.667) |
| **4. Anxiety and fear-related disorders (6B0)** | | | | | | |
| No | 262 (93.91) | 17 (6.09) | 0.58 (0.445) | 132 (89.80) | 15 (10.20) | 0.68 (0.410) |
| Yes | 9 (100) | 0 | | 6 (100) | 0 | |
| **5. Neurodevelopmental disorders (6A0)** | | | | | | |
| No | 264 (93.95) | 17 (6.05) | 0.45 (0.502) | 135 (90.00) | 15 (10.00) | 0.33 (0.564) |
| Yes | 7 (100) | 0 | | 3 (100) | 0 | |
| **6. Factitious disorders (6D5)** | | | | | | |
| No | 266 (93.99) | 17 (6.01) | 0.32 (0.572) | 137 (90.13) | 15 (9.87) | 0.11 (0.741) |
| Yes | 5 (100) | 0 | | 1 (100) | 0 | |
| **7. Elimination disorders (6C0)** | | | | | | |
| No | 270 (94.08) | 17 (5.92) | 0.06 (0.802) | 138 (90.20) | 15 (9.80) | NA |
| Yes | 1 (100) | 0 | | 0 | 0 | |
| **8. Personality disorder (8D1)** | | | | | | |
| No | 269 (94.06) | 17 (5.94) | 0.13 (0.722) | 138 (90.20) | 15 (9.80) | NA |
| Yes | 2 (100) | 0 | | 0 | 0 | |
| **9. Dissociative disorders (6B6)** | | | | | | |
| No | 270 (94.08) | 17 (5.92) | 0.06 (0.802) | 138 (90.20) | 15 (9.80) | NA |
| Yes | 1 (100) | 0 | | 0 | 0 | |
| **10. Disruptive behavior and dissocial disorders (6C9)** | | | | | | |
| No | 270 (94.08) | 17 (5.92) | 0.06 (0.802) | 137 (90.13) | 15 (9.87) | 0.11 (0.741) |
| Yes | 1 (100) | 0 | | 1 (100) | 0 | |
| **11. Catatonia** | | | | | | |
| No | 271 (94.10) | 17 (5.90) | NA | 137 (90.13) | 15 (9.87) | 0.11 (0.741) |
| Yes | 0 | 0 | | 1 (100) | 0 | |
| **12. Disorders specifically associated with stress disorders (6B4)** | | | | | | |
| No | 267 (94.01) | 17 (5.99) | 0.25 (0.614) | 133 (90.48) | 14 (9.52) | 0.33 (0.564) |
| Yes | 4 (100) | 0 | | 5 (83.33) | 1 (16.67) | |

(*Continued*)

**Table 2.** (Continued)

| Variables | Before COVID-19 pandemic | | | During the COVID-19 pandemic | | |
|---|---|---|---|---|---|---|
| | No | Yes | $\chi^2$ (*p*-value) | No | Yes | $\chi^2$ (*p*-value) |
| | (271, 94.10%) | (17, 5.90%) | | (138, 90.20%) | (15, 9.80%) | |
| 13. Mental and behavioral disorders associated with pregnancy, childbirth and the puerperium, (6E2) | | | | | | |
| No | 271 (94.10) | 17 (5.90) | NA | 137 (90.13) | 15 (9.87) | 0.11 (0.741) |
| Yes | 0 | 0 | | 1 (100) | 0 | |
| 14. Disorders of bodily distress and bodily experience (6C2) | | | | | | |
| No | 264 (93.95) | 17 (6.05) | 0.45 (0.502) | 136 (90.07) | 15 (9.93) | 0.22 (0.639) |
| Yes | 7 (100) | 0 | | 2 (100) | 0 | |
| 15. Impulsive control disorders (6C7) | | | | | | |
| No | 269 (94.06) | 17 (5.94) | 0.13 (0.722) | 136 (90.07) | 15 (9.93) | 0.22 (0.639) |
| Yes | 2 (100) | 0 | | 2 (100) | 0 | |
| 16. HIV | | | | | | |
| No | 261 (93.88) | 17 (6.12) | 0.65 (0.420) | 135 (90.00) | 15 (10.00) | 0.33 (0.564) |
| Yes | 10 (100) | 0 | | 3 (100) | 0 | |

### Factors associated with having a substance use disorder diagnosis

The variables that were significant (*p*<0.05) at inferential statistics (i.e., chi-square and t-test), were included in logistic analyses and those significant at bivariate logistic regressions were considered for model building. For both periods combined, all the included variables; age, comorbidity of schizophrenia and other primary psychosis (6A2), and bipolar disorder had a VIF below 3 and mean VIF of 1.21, indicating good collinearity. The final model had a sensitivity of 0%, specificity of 100%, a negative predictive value of 92.74%, and would correctly classify 92.74% of the diagnosis of SUD in both periods. The model had good goodness of fit for all the included variables, with a *p*-value of 0.590. The likelihood of being diagnosed with SUD increased with age [adjusted odds ratio (aOR) = 1.81, 95% confidence interval (CI) = 1.31–2.52, *p*<0.001]. However, having a diagnosis of bipolar mood disorder (aOR = 0.04, CI = 0.01–0.22, *p* = 0.001) and schizophrenia and other primary psychosis (aOR = 0.03, CI = 0.01–0.13, *p*<0.001) was protective against SUD.

*Before* the pandemic, only an increase in age was associated with SUD; whereas an increase in age and having comorbidity of bipolar disorder were associated with SUD *during* the pandemic. These were tested for collinearity and all had a VIF of 1. Therefore, they were included in the final model, which had a sensitivity of 0%, specificity of 100%, a negative predictive value of 90.20%, and would correctly classify 90.20% of the diagnosis of SUD *during* the pandemic. The likelihood of being diagnosed with SUD increased with an increase in age *during* the pandemic (aOR = 1.72, CI = 1.06–2.80, *p* = 0.028). However, having a diagnosis of bipolar disorder reduced the likelihood of being diagnosed with SUD (aOR = 0.18, CI = 0.04–0.87, *p* = 0.032) (Table 3).

### Discussion

This study aimed to investigate substance use disorder (SUD) among adolescents *before* and *during* the COVID-19 pandemic using a retrospective record review of the outpatient register (HMIS form 031 at the Mbarara Regional Referral psychiatry ward, the largest tertiary hospital in southwestern Uganda). The overall prevalence of SUD was 7.26% considering all the adolescent records over two periods (5.90% and 9.80%, *before* and *after* the pandemic, respectively). There was no statistically significant difference in SUD during the two periods despite a

**Table 3. Logistic analysis for factors associated with substance use disorder.**

| Variable | Total sample | | | | Before the pandemic | | During the pandemic | | | |
|---|---|---|---|---|---|---|---|---|---|---|
| | Bivariable analysis | | Multivariable analysis | | Bivariable analysis | | Bivariable analysis | | Multivariable analysis | |
| | Crude Odds ratio (95% confidence interval) | p-value | Crude Odds ratio (95% confidence interval) | p-value | Crude Odds ratio (95% confidence interval) | p-value | Crude Odds ratio (95% confidence interval) | p-value | Adjusted odds ratio (95% confidence interval) | P-value |
| **Age** | 1.86 (1.31–2.62) | <**0.001** | 1.81 (1.31–2.52) | <**0.001** | 1.98 (1.20–3.24) | **0.007** | 1.76 (1.08–2.86) | **0.023** | 1.72 (1.06–2.80) | **0.028** |
| **Gender** | | | | | | | | | | |
| Female | 1 | | | | 1 | | 1 | | | |
| Male | Omitted | | | | Omitted | | Omitted | | | |
| **Schizophrenia and other primary psychosis (6A2)** | | | | | | | | | | |
| No | 1 | | 1 | | 1 | | 1 | | | |
| Yes | 0.77 (0.01–0.57) | **0.012** | 0.03 (0.01–0.22) | **0.001** | Omitted | | 0.16 (0.02–1.28) | 0.085 | | |
| **Bipolar mood disorders** | | | | | | | | | | |
| No | 1 | | 1 | | 1 | | 1 | | 1 | |
| Yes | 0.09 (0.02–0.37) | **0.001** | 0.04 (0.01–0.17) | <**0.001** | Omitted | | 0.19 (0.04–0.87) | **0.032** | 0.18 (0.04–0.87) | **0.032** |

slightly higher rate reported *during* the pandemic. In addition, the use of cigarettes was statistically higher *during* the COVID-19 pandemic. The likelihood of being diagnosed with SUD was more among older adolescents at any period. However, having a diagnosis of bipolar mood disorder reduced the likelihood of being diagnosed with SUD *during* the pandemic.

Despite a little rising in SUD (3.9% increase) *during* the COVID-19 pandemic, there was no statistically significant change compared to *before* the pandemic; a finding similar to other studies that found that cannabis and alcohol use did not change during the pandemic [36]. Lockdowns during the pandemic presented an opportunity to reduce the availability of substances due to the disruption in the distribution chain, as seen in other countries. For example, a 17% and 24% decline in marijuana and alcohol availability in the USA was reported during the pandemic; given the situation of substance unavailability in the USA, the substance use rates had not significantly decreased [36]. Despite these findings, other studies reported increased substance use among adolescents during the pandemic of 25.33% and 9.33% increment in alcohol and other substance use, respectively [37].

The use of substances in our study was higher than that reported in Indonesia at 5.3% for the commonly used substance—alcohol [38]. Cigarette use was statistically significantly more during the COVID-19 pandemic, which is contradictory to other studies that found no significant comparison [36]. These findings indicate that adolescents procured and accessed substances from their own homes since schools were closed during the pandemic [39]. Vaping materials and cigarettes are generally obtained through peer networks [40], and their availability kept increasing due to increased interaction during school closure making cigarettes more accessible during the lockdown.

This study found that an increase in age was associated with an increased likelihood of being diagnosed with SUD, a factor consistently associated with substance use among adolescents, as reported by many researchers [38, 40]. An increase in age leads to increased curiosity, bandwagon effect, and impulsivity towards substance use [41]. The use of addictive substances, particularly alcohol, is restricted under 18 in Uganda [42]. Therefore, adolescents who tend towards the accepted age (late adolescence, 18 to 19 years in this study; for example) were more likely to be diagnosed with SUD.

Males are more likely than females to take alcohol and drugs. The more exposure to a substance, the higher the risk of being diagnosed with SUD. Male's predominance in substance

use may be associated with the cultural acceptance of substance use among the male gender, especially in this patriarchal society [43, 44]. Also, the use of substances shows the masculinity in an individual man since weak men are believed to be unable to handle alcohol or other substances [44, 45]. Moreover, adolescents are always watching television and on the internet, which has adverts promoting male use of substances. For example, the use of alcohol by most celebrities in songs and most sports are sponsored by alcohol-producing companies [28]. The increased likelihood of being diagnosed with SUD among males could also be an effect of the increased tolerance towards alcohol that puts them at a higher risk for SUD [45, 46].

Individuals with primary psychotic or mood disorders usually use addictive substances to cope with their symptoms [47, 48]. Although teenagers with primary psychosis or mood symptoms mostly experience their first episodes of the primary disorder, most of them have not yet developed coping mechanisms, including substance use, unlike their counterparts who have experienced several episodes. This may be the case supporting the present finding that having a diagnosis of a primary psychotic or mood disorder was less associated with SUD.

The interpretation of this study's findings should be made with caution in view of the following limitations. First of all, it was a retrospective cross-sectional study whose data depended on the efficiency of record reporting by the department staff, an aspect that has caused inconsistencies with HMIS reporting [49]. In addition, many variables that could have influenced the findings, such as stressors and family environment, were not retrieved from the registry and hence not included in the analysis. Finally, the data included in the analysis were from one facility, limiting the generalization of the findings throughout the country.

## Conclusions

The present study is one of the first approaches to investigating the change of SUD after the COVID-19 pandemic inception. The findings reported herein indicate a slight increment of SUD among Ugandan adolescents during the pandemic. Therefore, there is a need to screen and treat adolescents for SUD as the pandemic progresses, especially among male-older adolescents. Some strategies, for example, reopening schools, and increasing support in schools and families for managing and identifying SUD to contribute towards early intervention for preventing the hazardous consequences of SUD, such as an increase in crime, are highly suggested.

## Acknowledgments

The authors acknowledge their affiliation, the African Center for Suicide and Research. Also, acknowledge the support of the Department of Psychiatry Mbarara Regional Referral Hospital.

## Author Contributions

**Conceptualization:** Mark Mohan Kaggwa, Joan Abaatyo, Sarah Maria Najjuka, Scholastic Ashaba, Mohammed A. Mamun.

**Data curation:** Mark Mohan Kaggwa, Alain Favina, Mohammed A. Mamun.

**Formal analysis:** Mark Mohan Kaggwa, Joan Abaatyo, Moses Muwanguzi, Sarah Maria Najjuka, Mohammed A. Mamun.

**Funding acquisition:** Mark Mohan Kaggwa, Sarah Maria Najjuka.

**Investigation:** Mark Mohan Kaggwa, Joan Abaatyo, Emmanuel Alol, Moses Muwanguzi, Sarah Maria Najjuka, Scholastic Ashaba, Mohammed A. Mamun.

**Methodology:** Mark Mohan Kaggwa, Joan Abaatyo, Emmanuel Alol, Moses Muwanguzi, Alain Favina, Scholastic Ashaba, Mohammed A. Mamun.

**Project administration:** Mark Mohan Kaggwa, Joan Abaatyo, Godfrey Zari Rukundo, Mohammed A. Mamun.

**Resources:** Mark Mohan Kaggwa, Moses Muwanguzi, Alain Favina.

**Supervision:** Mark Mohan Kaggwa, Emmanuel Alol.

**Validation:** Moses Muwanguzi.

**Visualization:** Mark Mohan Kaggwa, Moses Muwanguzi, Sarah Maria Najjuka.

**Writing – original draft:** Mark Mohan Kaggwa, Mohammed A. Mamun.

**Writing – review & editing:** Mark Mohan Kaggwa, Joan Abaatyo, Emmanuel Alol, Moses Muwanguzi, Sarah Maria Najjuka, Alain Favina, Godfrey Zari Rukundo, Scholastic Ashaba, Mohammed A. Mamun.

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
