## [Decision Letter · Decision Letter 0]

2 May 2022

PONE-D-22-08744Substance use disorder among adolescents before and during the COVID-19 pandemic in Uganda: Retrospective findings from a psychiatric ward registryPLOS ONE

Dear Dr. Kaggwa,

Thank you for submitting your manuscript to PLOS ONE. After careful consideration, we feel that it has merit but does not fully meet PLOS ONE’s publication criteria as it currently stands. Therefore, we invite you to submit a revised version of the manuscript that addresses the points raised during the review process.

We look forward to receiving your revised manuscript.

Kind regards,

Felix Bongomin, MB ChB, MSc, MMed, FECMM

Academic Editor

PLOS ONE

Journal Requirements:

Additional Editor Comments:

Dear authors ,

We got favorable comments from the reviewer.

Please address the comments regarding adolescence population in Uganda and the title .

Reviewers' comments:

Reviewer's Responses to Questions

**Comments to the Author**

1. Is the manuscript technically sound, and do the data support the conclusions?

Reviewer #1: Yes

2. Has the statistical analysis been performed appropriately and rigorously? 

Reviewer #1: I Don't Know

3. Have the authors made all data underlying the findings in their manuscript fully available?

Reviewer #1: Yes

4. Is the manuscript presented in an intelligible fashion and written in standard English?

Reviewer #1: Yes

5. Review Comments to the Author

Reviewer #1: My only input is that I found the title of the manuscript not elaborately representative; this study does not describe Ugandan adolescence,, its more like a description of the adolescents who have mental disorders in Uganda

6. PLOS authors have the option to publish the peer review history of their article (what does this mean?). If published, this will include your full peer review and any attached files.

Reviewer #1: No

---

## [Author Response · Author response to Decision Letter 0]

11 May 2022

Comment: My only input is that I found the title of the manuscript not elaborately representative; this study does not describe Ugandan adolescence, its more like a description of the adolescents who have mental disorders in Uganda.

Response: We are thankful to the reviewer for their time to read and understand our manuscript. This study was a snapshot of adolescents diagnosed with substance use disorder during the COVID-19 pandemic and not necessarily those who had an only mental illness. This is also reflected in our study findings; that is, only five adolescents had a comorbid mental illness (i.e., depression, schizophrenia, etc.). Many adolescents used substances of addiction, but the study was interested in capturing those who were admitted due to complications of the substance use, i.e., substance use disorders

---

## [Editor Report · Decision Letter 1]

13 May 2022

Substance use disorder among adolescents before and during the COVID-19 pandemic in Uganda: Retrospective findings from a psychiatric ward registry

PONE-D-22-08744R1

Dear Dr. Kaggwa,

We’re pleased to inform you that your manuscript has been judged scientifically suitable for publication and will be formally accepted for publication once it meets all outstanding technical requirements.

Kind regards,

Felix Bongomin, MB ChB, MSc, MMed, FECMM

Academic Editor

PLOS ONE
---

## [Editor Report · Acceptance letter]

17 May 2022

PONE-D-22-08744R1 

Substance use disorder among adolescents before and during the COVID-19 pandemic in Uganda: Retrospective findings from a psychiatric ward registry 

Dear Dr. Kaggwa:

I'm pleased to inform you that your manuscript has been deemed suitable for publication in PLOS ONE. Congratulations! Your manuscript is now with our production department. 

Kind regards, 

on behalf of

Dr. Felix Bongomin 

Academic Editor

PLOS ONE